# Asynchrony of *Gambierdiscus* spp. Abundance and Toxicity in the U.S. Virgin Islands: Implications for Monitoring and Management of Ciguatera

**DOI:** 10.3390/toxins13060413

**Published:** 2021-06-10

**Authors:** Justin D. Liefer, Mindy L. Richlen, Tyler B. Smith, Jennifer L. DeBose, Yixiao Xu, Donald M. Anderson, Alison Robertson

**Affiliations:** 1Department of Biology, Mount Allison University, New Brunswick, NJ E4L 1E4, Canada; jliefer@mta.ca; 2U.S. Food and Drug Administration, Dauphin Island, Alabama, AL 36528, USA; 3Woods Hole Oceanographic Institute, Woods Hole, MA 02543, USA; mrichlen@whoi.edu (M.L.R.); xuyixiao_77@163.com (Y.X.); danderson@whoi.edu (D.M.A.); 4Center for Marine and Environmental Studies, University of the Virgin Islands, St. Thomas, MN 00802, USA; tsmith@uvi.edu; 5School of Marine and Environmental Sciences, University of South Alabama, Mobile, Alabama, AL 36688, USA; jdebose@southalabama.edu; 6Key Laboratory of Environment Change and Resources Use in Beibu Gulf, Ministry of Education, Nanning Normal University, Nanning 530001, China; 7Dauphin Island Sea Lab., Dauphin Island, Alabama, AL 36528, USA

**Keywords:** *Gambierdiscus*, ciguatera poisoning, *Dictyota*, ciguatoxin, Caribbean, dinoflagellate, benthic algae, algal toxin, harmful algal bloom

## Abstract

Ciguatera poisoning (CP) poses a significant threat to ecosystem services and fishery resources in coastal communities. The CP-causative ciguatoxins (CTXs) are produced by benthic dinoflagellates including *Gambierdiscus* and *Fukuyoa* spp., and enter reef food webs via grazing on macroalgal substrates. In this study, we report on a 3-year monthly time series in St. Thomas, US Virgin Islands where *Gambierdiscus* spp. abundance and Caribbean-CTX toxicity in benthic samples were compared to key environmental factors, including temperature, salinity, nutrients, benthic cover, and physical data. We found that peak *Gambierdiscus* abundance occurred in summer while CTX-specific toxicity peaked in cooler months (February–May) when the mean water temperatures were approximately 26–28 °C. These trends were most evident at deeper offshore sites where macroalgal cover was highest year-round. Other environmental parameters were not correlated with the CTX variability observed over time. The asynchrony between *Gambierdiscus* spp. abundance and toxicity reflects potential differences in toxin cell quotas among *Gambierdiscus* species with concomitant variability in their abundances throughout the year. These results have significant implications for monitoring and management of benthic harmful algal blooms and highlights potential seasonal and highly-localized pulses in reef toxin loads that may be transferred to higher trophic levels.

## 1. Introduction

Of all the human poisoning syndromes associated with harmful algal blooms (HABs), ciguatera poisoning (CP) has the most significant human health and economic impacts globally [1]. CP is caused by consumption of fish or shellfish, generally associated with coral reef systems, that are contaminated with a suite of lipid-soluble toxins known collectively as ciguatoxins. These toxins and their precursors are produced by certain species or strains of benthic dinoflagellates in the genera *Gambierdiscus* and *Fukuyoa* that live on algal substrates or other surfaces, such as dead corals and sand, in many coral reef communities [2,3,4]. Ciguatoxins (CTX) and related metabolites enter and accumulate in coral reef food webs through grazing by herbivorous fish and invertebrates, reaching their highest concentrations in carnivorous finfish [5,6,7], where they pose the greatest public health risk [8]. With increased globalization, CTX and *Gambierdiscus* have been reported from temperate locations including the northern Gulf of Mexico, New Zealand, Japan, and the Canary Islands; however, CP is endemic to many tropical and subtropical coral reef ecosystems globally and primarily affects coastal communities [9,10,11,12]. Global estimates of CP incidence vary widely, ranging from tens of thousands to as many as 500,000 poisonings per year [1,13]. Producing accurate estimates of the true incidence of CP is challenged by a high degree of underreporting and misdiagnoses [14], and consequently, CP remains an overlooked and under-appreciated problem. Prevention and management strategies for CP have been hindered by knowledge gaps regarding the environmental and physiological factors contributing to toxin dynamics, as well as the lack of commercially available toxin standards and affordable and practical methods for toxin detection.

CP differs from other HABs in that poisoning events are not associated with large-scale planktonic blooms of a single causative species but are often an ongoing and chronic problem in endemic regions. Over the past two decades, renewed scientific interest and research has resulted in significant advances in our understanding of the biogeography and ecophysiology of *Gambierdiscus* and *Fukuyoa,* including a fuller characterization of species diversity and global distribution [2,15], intra- and inter-specific [16,17,18,19] growth characteristics [20,21], and habitat or substrate preferences (reviewed by [2]). These studies have provided a fuller understanding of the factors governing population and toxin dynamics, including the identification of highly toxic species that may dominate CTX production and flux into food webs. Key drivers of CTX prevalence and CP risk are thought to involve a combination of several environmental and ecological factors, including: (1) environmental conditions that promote growth, leading to high *Gambierdiscus* and *Fukuyoa* cell concentrations, (2) prevalence of CTX-producing species and strains, (3) environmental conditions that promote cellular toxin production, and (4) increased substrate availability that promotes the proliferation and increased areal abundance of benthic dinoflagellates in reef ecosystems. Additionally, patterns of toxicity are affected by bioconversion of the toxin precursors produced by *Gambierdiscus* to more potent compounds during toxin uptake, metabolism, and transfer.

As research on *Gambierdiscus* has expanded and progressed, so have efforts to characterize linkages between key environmental factors (e.g., seawater temperature), *Gambierdiscus* and *Fukuyoa* population and toxin dynamics, and subsequent CP risk and incidence. Early efforts by Tosteson [22] reported a relationship between warming seawater temperatures and barracuda toxicity, in which barracuda ciguatoxicity was correlated with both increases in seawater temperatures as well as with total cases of human ciguatera intoxications. In Tahiti, Chinain et al. [23] carried out weekly sampling of *Gambierdiscus* abundance and toxicity over a seven-year period (1993–1999), which identified seasonal trends in peak cell densities that occurred during the warmest months (October, November, December). They concluded that ciguatera outbreaks more likely reflected the presence of highly toxic strains rather than high overall biomass, as no correlation was found between sample toxicity and *Gambierdiscus* spp. abundance. These data were subsequently used by Chateau-Degat et al. [24] to construct a temporal model that related seawater temperatures to *Gambierdiscus* spp. growth, and subsequent onset of ciguatera cases. A key challenge in these and other ongoing efforts to link spatiotemporal dynamics of *Gambierdiscus* spp. abundance and CTX production with CP risk is the high spatial heterogeneity observed for both *Gambierdiscus/Fukuyoa* populations and the ciguatoxicity of potential seafood vectors. Within small spatial scales (<3 km) on a single island, fish at one reef site may be safe to eat, while neighboring reefs can harbor “hot-spots” where ciguatoxic fish are prevalent [25,26,27,28]. This variation in CTX accumulation within higher trophic levels may reflect corresponding spatial heterogeneity in *Gambierdiscus* population structure, coupled with the large differences in CTX production documented among co-occurring *Gambierdiscus* and *Fukuyoa* species [9,16,17,29,30]. For example, there can be an over 1500-fold difference in toxin content among *Gambierdiscus* species, with *G. polynesiensis* identified as a toxic species from the Pacific [7], and *G. excentricus* and *G. silvae* as the most toxic species from the Caribbean [17,19,31,32].

In this study, we assessed *Gambierdiscus* abundance and CTX content within natural epiphyte assemblages from St. Thomas, USVI (Figure 1), an area hyperendemic for CP [33], to determine the seasonality, spatial variability, and environmental correlates of CTX production. Field activities were conducted monthly over 3 years at four coral reef sites located on the south side of the island, including two nearshore (Black Point, BP; Coculus Rock, CRK) and two offshore locations (Flat Cay, FC; Seahorse Shoal, SH), ranging in depth from ~6–22 m depth (see Figure 1). Populations of *Gambierdiscus* found at these sites are known to comprise five of the seven *Gambierdiscus* species documented in the Caribbean and one ribotype: *G. belizeanus*, *G. caribaeus*, *G. carolinianus*, *G. carpenteri*, *G. silvae,* and *G.* sp, *ribotype 2* (Richlen, M.L. unpublished data, and [34]). Clear, yet decoupled, seasonal patterns of *Gambierdiscus* abundance and CTX levels were observed, as well as marked differences in CTX levels among adjacent sites. Our findings indicate that variability in CTX production within *Gambierdiscus* populations over small spatial scales may be a key driver of CP risk. This work also highlights the importance of time-integrated monitoring of in situ CTX production, which provides a more direct determination of the sites and conditions that are the ultimate source of CP risk.

## 2. Results

### 2.1. Variation in Environmental Conditions and Benthic Composition

Mean daily temperature and salinity (not shown) were similar at all four sites and followed seasonal patterns. Benthic temperatures ranged from 25.5–29.9 °C with a mean of 27.8 ± 1.6 °C and displayed a seasonal pattern typical of the tropical northern hemisphere, with peak temperatures in summer (June–October) and minimum temperatures in winter (December–March). Salinity variations were small and ranged from 34.6–36.2 psu with a mean of 35.5 ± 0.5 psu, with maximum values in March–May and minimum values in September–November. Dissolved nutrients were low overall and varied over a small range. For instance, mean dissolved inorganic phosphorus (DIP) was 0.09 ± 0.04 μM, while mean dissolved inorganic nitrogen was 0.90 ± 0.8 μM. There were also no strong seasonal patterns in available data on climate variables (precipitation, wind speed and direction). Spatial and seasonal variation in environmental and physical parameters (i.e., wind speed and direction, precipitation, benthic temperature, and nutrients) were also examined with multivariate techniques and no clear ordination of parameters or clustering of sites was apparent (see Appendix A). Analyses were limited to these parameters due to frequent gaps in other data (e.g., salinity and other CTD vertical profile measurements). Selected environmental parameters were compared using a principal components analysis (PCA) and no principal components explained more than 22.8% of overall variation (Appendix A). Environmental variation among sites was examined with a cluster analysis, multidimensional scaling, and an ANOSIM test (Primer-E) for parameters that were measured directly at each site (benthic temperature and nutrients). All sites showed high multivariate similarity (ANOSIM Global R = 0.28) and no clear spatial patterns.

Benthic composition at all sites was mostly dominated by macroalgae, which ranged from 11.7–75.0% of benthic cover with a mean of 39.6 ± 12.5%, and was variable over time (Figure 2). Other major components were dead coral with turf algae, non-living substrate, and corals, with mean benthic cover ranging from 11.7–23.0%. Minor components included gorgonians, sponges, coralline algae, and cyanobacteria. As with the site environmental conditions, there were no clear spatiotemporal patterns and a high multivariate similarity (ANOSIM test, Primer-E) in overall benthic cover among sample years and study sites. Macroalgal cover was dominated by fleshy macroalgae which mainly consisted of *Dictyota* spp. (24.1 ± 10.0% of macroalgal cover; see monthly trends in Figure 2), followed by *Lobophora variegata* (7.7 ± 10.4%), and *Halimeda* spp. (0.4 ± 1.2%). Macroalgae composition was distinct among the sites (ANOSIM Global R = 0.568) (Appendix A), primarily due to differences between the nearshore sites (CRK and BP) and the offshore Sites FC (R = 0.501–0.539) and SH (R = 0.824–0.935) (Appendix A). A SIMPER analysis showed these dissimilarities were driven by the higher abundance of *L. variegata* at the offshore sites, which accounted for 41.7–60.7% of the dissimilarity between pairwise comparisons.

### 2.2. Gambierdiscus spp. Abundance

*Gambierdiscus* spp. were detected in all 135 samples collected in this study. Abundances of *Gambierdiscus* were highly variable, both in terms of overall range (2.5–63.3 cells g *Dictyota*^−1^, mean 69.0 ± 63.3 cells g *Dictyota*^−1^) and periodicity (Figure 3). The only apparent seasonal pattern in abundance was the occurrence of an annual maximum in September–October of each study year at each site that coincided with the thermal maximum of sea surface temperature and doldrum-like conditions. The variation in *Gambierdiscus* spp. abundance was dissimilar among sites, with the exception of the annual abundance peaks in September–October of 2010 and 2012 (see Figure 3). The highest overall abundances at CRK, FC, and SH were observed in September–October 2010, and annual mean *Gambierdiscus* spp. abundance was also significantly higher (*p* < 0.001, ANOVA with Tukey’s test post hoc) in 2010 (111.5 ± 80.8 cells g *Dictyota*^−1^) compared to 2011 and 2012 (49.4 ± 35.6 and 52.7 ± 51.3 cells g *Dictyota*^−1^, respectively). Mean abundances were generally higher at BP (84.7 ± 69.3 cells g *Dictyota*^−1^) and FC (80.4 ± 56.9 cells g *Dictyota*^−1^) near the western end of St. Thomas compared to the eastern sites of CRK and SH (56.6 ± 62.0 and 54.7 ± 61.6 cells g *Dictyota*^−1^, respectively), with overall abundance at BP being significantly higher than at CRK and SH (*p* < 0.05). There was no significant correlation between abundance and any of the environmental factors assessed, based on both direct parametric correlation tests and multivariate correlation analysis (BEST in Primer-E).

### 2.3. Detection of Ciguatoxins in Field Samples

The specific sodium channel agonist activity detected by N2a assay was attributed to CTX congeners in all samples based on several lines of evidence: (1) the direction, shape, and slope of dose-response curves generated from field sample extracts were congruent with C-CTX-1 standards indicating similar activities and potencies by N2a assay (see Appendix A); (2) *Gambierdiscus* spp. (a known source of CTXs) was present in all samples; (3) the extraction procedure used was not suitable for isolation of the polar alkaloid sodium channel blockers known to occur in marine systems (e.g., saxitoxin, tetrodotoxin); (4) non-specific activity potentially generated by other toxin classes (with alternate modes of action) were excluded from our analyses (as described in the Methods); and, (5) dinoflagellate sources and toxins of other site 5 sodium channel agonists, e.g., *Karenia*-produced brevetoxins, have not been reported in algae, fish, or shellfish from the study region. Considering the biotransformation of CTXs characterized in other regions, the CTX congeners detected in Caribbean *Gambierdiscus* spp. are likely to be uncharacterized precursors of C-CTX-1 or C-CTX-2, the most abundant congeners found in higher trophic level Caribbean fish [37,38]. The Caribbean CTX standard used in the N2a bioassays was C-CTX-1, the only quantified reference material that was available at the time of this study, hence all detected CTX levels are expressed as C-CTX-1 equivalents (C-CTX eq.).

The identity, structure, and toxicity of Caribbean CTX congeners present in *Gambierdiscus* spp. are poorly understood and no reference materials are presently available. This lack of knowledge complicates the use of clean-up methods, such as solid-phase extraction (SPE), for sample extracts as they may remove the target analytes. As detailed in the methods, four samples, representing a range of determined CTX activities, were purified using silica (Si) SPE column (Agilent) to assess the effect of sample clean up on measured composite toxicity. In all four samples, Si SPE purification caused a reduction in assay response in both the ouabain-veratridine-treated cells (i.e., reduction in CTXs) and PBS control cells (i.e., a reduction in cytotoxic matrix compounds). This indicates that Si SPE clean-up may have removed cytotoxic matrix compounds that affect negative control N2a cells, but also removed some of the target analyte with no improvement in quantification and hence was not used for sample quantitation. Dilution of samples (reducing both interfering matrix and analyte) that did not undergo clean-up or purification resulted in a 7–16% variation in quantitation. Considering the lack of precision in cell-based and other bioassays, as compared to instrumental methods (e.g., LC-MS/MS), this was considered an acceptable degree of variation.

CTX activity was quantifiable in 24.6% of benthic algal samples without purification while also meeting quality assessment controls of the N2a assay (summarized in Table 1). In the vast majority of remaining samples, CTX activity was below the limit of quantitation. The lowest CTX concentration quantified in our samples was 0.33 ± 0.06 ng C-CTX-1 eq. mL^−1^, which was above the determined limit of quantitation for CTX in unpurified extracts of environmental algal samples using the N2a assay (see Section 4.4.4).

### 2.4. Spatial and Temporal Variability of In Situ Ciguatoxin Levels

Among the 135 sampling events, 125 samples were available for assessing CTX production by *Gambierdiscus*. Among these, 46 (37.7%) were positive for CTX activity (Table 1), meaning that there was ≤50% survival in ouabain-veratridine-treated (i.e., sensitized to sodium channel agonist) N2a cells and ≥95% survival in untreated N2a cells at the same dosage. CTX levels were high enough to be quantified in 30 (24.6%) samples (Table 1). Samples deemed positive for CTX (specific activity for a sodium channel agonist), but not meeting quantitation criteria, were considered “trace” detections. Three samples were also considered trace measurements due to detection of CTX in the toxin samples and a corresponding detection of *Gambierdiscus* spp. in the abundance samples at the same sampling points, but a measurement of *Gambierdiscus* cell abundance in the samples collected to assess toxicity was not available. Cell toxin quotas for *Gambierdiscus* spp. were calculated by normalizing the measured CTX concentration to the *Gambierdiscus* cell abundance measured within a given toxin sample. The toxin load for each sample was calculated as the product of the cell toxin quota and the *Gambierdiscus* spp. abundance measured on *Dictyota* hosts (i.e., cells per g *Dictyota*) during the same sample collection. This toxin load represents the amount of toxin present per mass of macroalgal substrate (units of pg C-CTX-1 eq. g *Dictyota*^−1^) and is used as a proxy for the amount of toxin available for trophic transfer during each sampling event.

The toxin quota of *Gambierdiscus* ranged from 0–12.6 pg C-CTX-1 eq. cell^−1^ (Table 1; Figure 4) and toxin load ranged from 0–453.8 pg C-CTX-1 eq. g *Dictyota*^−1^ (Table 1; Figure 3). Unlike *Gambierdiscus* spp. abundance, both toxin quota and toxin load appeared to have a distinct annual seasonality. The majority of positive samples (80.0%) and quantifiable toxin samples (76.6%) were collected during February–June of each sample year (Figure 3 and Figure 4). The six quantifiable toxin samples observed outside of this February–June season were collected from Sites FC and SH in July–August 2011 and January–February 2012, months adjacent to the greatest periods of toxin occurrence (February–June 2011 and 2012). Only four trace toxin detections were observed in September–November of all sample years. There were no significant differences in mean toxin quota or toxin load between sample years (*p* = 0.057–0.061), although the largest proportion of quantifiable toxin samples occurred in 2012 (50.0%) followed by 2011 (33.3%) and 2010 (16.7%).

The majority of positive (67.2%) and quantifiable samples (76.7%) originated from the offshore sites FC and SH, with 46.7% of quantifiable samples originating from site FC alone and 30% from site SH (Table 1). The highest toxin quotas were detected at nearshore sites CRK (12.6 pg C-CTX-1 eq. cell^−1^) and BP (8.4 pg C-CTX-1 eq. cell^−1^), though these values were far higher than the other, infrequent toxin detections at these sites (Figure 4). A Welch’s ANOVA (a one-way ANOVA that assumes unequal variance) showed a significant difference in toxin quota (*p* < 0.005) and toxin load (*p* < 0.01) among sites. A post hoc Games–Howell test showed that both toxin quota (*p* < 0.01) and toxin load (*p* < 0.05–0.01) were significantly higher at FC compared to CRK and BP (Figure 5). Considering both spatial and temporal variability, 50.0% of positive samples and 63.3% of quantified samples occurred at offshore sites and during the high toxicity period of February–June.

There was no strong or significant correlation between toxin quota or toxin load and any of the environmental factors assessed, based on both direct parametric correlation tests and multivariate correlation analysis (BEST in Primer-E). This is not surprising considering that CTX was not quantifiable in ~75% of samples and the lack of ordination or apparent structure in the environmental data. Although no monitored environmental variables were directly correlated with toxin parameters, the vast majority of positive (69.6%) and quantifiable (90.0%) samples were collected when temperatures were below the mean of the study period (27.8 ± 1.6 °C). Additionally, greater majorities of positive (91.3%) and quantifiable (96.7%) samples were collected when salinity was above the mean of the study period (35.5 ± 0.5 °C).

The toxin quota of the *Gambierdiscus* present, rather than *Gambierdiscus* abundance, appeared to determine toxin load throughout the study. There was no correlation between *Gambierdiscus* abundance (Spearman’s R = −0.090, *p* = 0.51) and toxin load overall (Spearman’s R = −0.054, *p* = 0.68) Additionally, there were only four trace detections during the months of peak abundance in each year (September–November), while some of the highest toxin quotas and loads occurred when abundances were relatively low (Figure 3).

## 3. Discussion

Laboratory studies of isolated toxic microalgae are essential for confirming their toxicity and mechanisms of toxin production, but these measurements often differ from the levels of toxin production observed in natural systems. Thus, robust assessments of in situ toxin production by harmful microalgae are critical for understanding their true toxin dynamics and potential threat to ecosystem function or public health. This work is the most comprehensive and quantitative assessment to date of the in situ toxicity of *Gambierdiscus* in the Caribbean, the ultimate cause of more cases of human illnesses than any other harmful alga [1]. We also express this in situ toxicity in proportion to the mass of an abundant benthic substrate (the macroalga *Dictyota* spp.) consumed by potential vectors of CTX, providing a quantitative link between toxin production and trophic transfer. As these values were determined in a location where CP is endemic with monthly sampling over 3 years, our findings provide valuable constraints for efforts to model in situ CTX levels and the trophic transfer of CTX in the Greater Caribbean region.

### 3.1. Relative Toxicity of In Situ Gambierdiscus

The cell toxin quotas determined in this study, ranging from 0–12.62 pg C-CTX-1 eq. cell^−1^ with a mean of 0.5 ± 1.63 pg C-CTX-1 eq. cell^−1^, are comparable to the limited number of other available in situ values (see Table 2). Values in Table 2 that were determined with the mouse bioassay, originally compiled as mouse units by Litaker et al. [15], were converted to composite CTX toxin quotas by assuming that one mouse unit is equivalent to 18 ng of CTX3C for Pacific samples and 72 ng of C-CTX-1 for Caribbean samples [39,40]. Though comparable to other in situ values, the toxin quotas we observed are also considerably higher than those of most cultured *Gambierdiscus* strains of Caribbean, eastern Atlantic, or Pacific origin measured using similar applications of the N2a assay as used in this study [17,19,41]. Most of these N2a toxin quotas were determined as CTX3C equivalents [17,19], a common Pacific congener of CTX that has been reported to be 2-fold more toxic than the Caribbean congener we used as a reference standard, C-CTX-1 [42]. This difference in standard toxicity could cause a lower CTX content to be determined in Caribbean strains measured with CTX3C as a standard rather than C-CTX-1. Even when taking this difference into account, the only Caribbean strains with toxin quota values comparable to the in situ values measured in this study are those from the species *G. silvae* (2.1–4.8 pg C-CTX-1 eq. cell^−1^ [31]) and *G. excentricus* (0.47 pg CTX3C eq. cell^−1^ and 1.43 pg CTX3C eq. cell^−1^, respectively [17,19]).

### 3.2. Cellular CTX Quota and Not Gambierdiscus Abundance Determines CTX Production

One of the most notable findings of this study is the greater influence of toxin quota rather than *Gambierdiscus* abundance on CTX load overall and within any given site. The vast majority of CTX detections occurred from February to June of each study year when *Gambierdiscus* abundance was relatively low, while CTX was generally not detected during periods with the highest *Gambierdiscus* abundances. The only other study to our knowledge that has monitored site-specific *Gambierdiscus* abundance and in situ toxicity over time [23] found a similar asynchrony between abundance and toxicity, with the highest in situ toxin measurements observed at relatively low *Gambierdiscus* abundance, and *Gambierdiscus* abundance being a poor predictor of in situ toxicity. This decoupling or asynchrony of *Gambierdiscus* abundance and CTX production is consistent with wide variation in toxicity that has been observed among *Gambierdiscus* species in all regions where the genus is endemic [7,17,19,30,32]. The far greater variation in toxicity among species compared to the variation within a species [19,41,52,53,54] indicates that CTX source levels are mostly determined by species composition rather than *Gambierdiscus* abundance at the genus level [17]. Additionally, the species that appear to be most common and widespread in the Caribbean, such as *G. caribaeus*, *G. carolinianus, G. belizeanus*, and *G. carpenteri*, have low toxicities [17,19], thus periods when these species are abundant may not be expected to result in high CTX levels. In contrast, Caribbean species such as *G. silvae* have been shown through our prior efforts to have high toxin quotas [31], while others have reported similar trends for *G. excentricus* [17,19]. In both cases, the reported cell toxin quotas reported in these high toxin-producing strains of the respective species, could have the capacity to produce the in situ CTX loads observed in this study, even at low abundances. The discovery of “super-producing” strains of *G. polynesiensis* in Pacific waters has generated the hypothesis that a small relative abundance of highly toxic *Gambierdiscus* species may dominate CTX production that leads to CP outbreaks [7,40,48]. The high CTX loads observed at relatively low in situ *Gambierdiscus* abundances, the high toxicity of some *Gambierdiscus* species, and the low toxicity of the most common *Gambierdiscus* species in our study region all support the hypothesis that highly toxic, low-abundance species of *Gambierdiscus* dominate CTX production in the Caribbean.

### 3.3. Nearshore vs. Offshore Sites

*Gambierdiscus* CTX production also displayed spatial patterns that were generally consistent across three years of monitoring. *Gambierdiscus* was present in all samples collected over the study period, yet the majority of samples containing quantifiable CTX were collected at offshore sites (FC and SH) during February–June. Though distinct in their contribution to regional CTX levels, sites FC and SH were similar to nearshore sites with respect to measured physical and chemical features. Despite these apparent similarities, depth may be a factor that distinguishes the high CTX offshore sites (FC and SH), where sampling depths were greater (~18 m bottom depth) than at the nearshore sites CRK and BP (~9 m bottom depth). This difference in depth may generate distinct light and water motion conditions for epiphytic *Gambierdiscus* populations between offshore and nearshore sites, the measurement of which were beyond the scope of this work. Measurements of light availability or optical qualities of the overlying water column were not available for the samples used in this study and physical conditions at all study sites were inferred from current and wave activity at one nearby buoy-monitoring location. However, water motion has been shown to be low, with little effect on *Gambierdiscus* populations in this study area [55], and is considerably lower than at locations where depth and water motion seem to affect *Gambierdiscus* abundance within epiphytic communities [56]. There is limited and conflicting evidence as to the physiological effect of light on toxin production within *Gambierdiscus* species or strains [44,57], but there are distinct growth-irradiance responses among co-occurring Caribbean *Gambierdiscus* species [20,21]. More favorable light conditions for constitutively more toxic species at deeper offshore sites could support the observed spatial patterns in CTX production. Future studies could test this supposition by determining if the most toxic species found at the offshore locations in this study or similar locations have lower optimal or maximum growth irradiances or are better adapted to the likely lower variability in irradiance of deeper benthic habitats.

Nearshore and offshore sites also varied in terms of the relative abundance and composition of macroalgae substrate. At the nearshore sites (BP and CRK), *Dictyota* spp. made up the majority of the macroalgal substrate for *Gambierdiscus* attachment. However, at the offshore sites (FC and SH), there was generally a higher percent-cover of fleshy macroalgae, as well as a greater proportion of species other than *Dictyota* present (e.g., *Lobophora variegata*). Although, beyond this study, to differentiate, there are multiple aspects where macroalgal abundance and composition might impact *Gambierdiscus* abundance and ciguatoxin transferability. Specifically, *Gambierdiscus*–macroalgae host interactions, which can be species-specific, might depend on shading potential, chemical cues, and host-palatability to higher trophic levels [58,59,60]. Since *Gambierdiscus* species determination has further developed since this study, future studies could determine if there are particular in situ host associations or macroalgal abundance that favor particularly toxic species/strains and if these associations are linked to the environmental conditions and benthic community compositions that vary between these nearshore and offshore sites.

### 3.4. Seasonality of CTX Production

Seasonal patterns of *Gambierdiscus* CTX quota and CTX loads were observed, with the vast majority (80.0%) of CTX detections occurring in February–June, which provides key insights into the environmental and ecological factors controlling CTX exposure risk. A similar seasonal pattern for in situ toxicity was also observed by Chinain et al. [23] in Tahiti, with the majority of high in situ CTX levels observed at temperatures below annual means and low or no in situ toxicity observed in the warmest months. If CTX loads are indeed determined by highly toxic species occurring at relatively low abundances, then the temporal patterns observed in this study indicate that these species show seasonality in their occurrence. Of the wide set of monitored environmental conditions that may affect *Gambierdiscus* populations, temperature and salinity showed the strongest seasonal patterns. However, neither of these factors were directly correlated with CTX quotas or loads. Salinity varied over a relatively small range (34.6–36.2 psu), which is unlikely to have an effect among species or a physiological impact within a species [20]. Temperature showed a considerably larger seasonal variation (25.5–29.9 °C) that spans the known range of temperature optima for *Gambierdiscus* species [20,21]. Despite the lack of a direct correlation between temperature and CTX levels, it is striking that the vast majority of positive (69.6%) and quantifiable (90.0%) samples were detected when benthic temperatures were below the mean temperature of the study area (27.8 ± 1.6 °C). This mean temperature is also well above the growth optimum for *G. silvae*, the most toxic Caribbean species examined by Xu et al. [20], which had the lowest upper temperature limit for growth (29.8 °C) among eight *Gambierdiscus* species. Additionally, the strain of *G. excentricus* that has produced the highest CTX quotas for an Atlantic species to date [19] was isolated from waters off the Canary Islands with relatively cool temperatures for *Gambierdiscus* (18–24 °C). The strain of *G. excentricus* that has produced the highest CTX quotas for a Caribbean species [17] was isolated from Pulley Ridge, located ~150 km offshore of Florida at a depth of 60–80 m where temperatures would be considerably lower than the mean temperature observed in the present study. The adaptation of the most toxic Caribbean strains to cooler temperatures is consistent with CTX detections being restricted to below-average temperatures in this study and may provide a key environmental constraint on CTX exposure risk.

### 3.5. Implications for Assessing CTX Exposure Risk

Our observation that CTX source levels in St. Thomas are determined by the toxin quota of *Gambierdiscus* cells rather than their abundance at the genus level has implications for efforts to predict CP risk in regions where this illness is endemic. Many proposed management efforts or models of potential CP risk are based on monitoring or predicting overall *Gambierdiscus* abundance [24,61,62,63]. Our findings and the apparent importance of species composition in determining the CTX production of a *Gambierdiscus* population [7,17] indicate that monitoring *Gambierdiscus* abundance alone would not help determine when and where trophic systems are likely to encounter and biomagnify CTX. Determining the most toxic *Gambierdiscus* species in an endemic CP location and using new molecular identification tools [34,64,65] to determine their spatiotemporal distribution may be more conducive to estimating CP exposure risk.

Even if the occurrence of CTX in the first trophic level could be accurately predicted in systems that yield ciguatoxic fish, predicting the spatial and temporal links between algal CTX production, bioaccumulation of CTX in higher trophic levels, and potential human exposure remain challenging. Our findings provide the basis for linking these phenomena by demonstrating that CTX production is restricted both spatially, within a relatively small study area (offshore sites in St. Thomas), and seasonally (~February–June). Determining the locations most likely to produce CTX allows studies of trophic dynamics of CTX (e.g., [26]) or of the site fidelity of key CTX vectors like large, mobile fish species to be related to a limited spatial source of CTX. By establishing a time-frame when CTX vectors are most likely to consume highly toxic *Gambierdiscus,* the lag between CTX production and potential human exposure can be better assessed. The restriction of CTX production to below-average temperatures and the possible importance of cool-adapted highly toxic species [8] also suggests lower temperature limits and a broader potential geographic range for CP risk in shallow marine habitats than previously estimated [63]. These implications (i.e., potential range expansion of cool-adapted toxigenic *Gambierdiscus* species) have also been suggested by others in the field [8,66,67], highlighting the importance of further evaluation so that monitoring and predictive models meet the needs of future risk assessment.

## 4. Materials and Methods

### 4.1. Site Descriptions

Samples were collected at four sites around St. Thomas (Figure 1) between late February/early March 2010 and December 2012. All St. Thomas sites are located south of the island on a nearshore to offshore gradient. Coculus Rock (CRK; 18.31257 N, 64.86058 W) is located near an emergent rock reef and is composed of diverse scattered stony corals on bedrock (6–7 m depth). Black Point (BP; 18.3445 N, 64.98595 W) is a nearshore fringing coral reef (7–16 m depth). Flat Cay (FC; 18.31822 N, 64.99104 W) is a fringing coral reef on the leeward side of a small uninhabited island (11–16 m depth). Seahorse Shoal (SH; 18.29467 N, 64.8675 W) is a deep patch reef 2 km offshore of St. Thomas (19–22 m depth). The latter three sites are star coral (*Orbicella* spp.) reefs with diverse coral and sponge communities. Further site descriptions can be found in [68]. During the sampling period for this study, these sites were impacted by a moderate thermal stress and coral bleaching event in August 2010 (widespread colony paling and bleaching, but limited mortality), which was truncated by the passage of Hurricane Earl on 3 August [69,70]. This storm passed about 105 km NE of St. Thomas, causing wind gusts of up to 120 km hr^−1^ and rainfall of 7.6 cm at the St. Thomas airport (see https://www.nhc.noaa.gov/data/tcr/AL072010_Earl.pdf, accessed on 22 April 2021).

### 4.2. Environmental Sampling

#### 4.2.1. Oceanographic Measurements

Salinity measurements were obtained at each site from vertical profiles taken with a shallow-water Seabird SBE 25 recording at 8 Hz (Sea-Bird Electronics, Bellevue, WA, USA). Sensors were factory-calibrated within one year of deployment. Casts were made at anchor or on drift within 100 m (horizontal) of the research site. Casts were made within 1 m of the seafloor. Resulting data files were trimmed to the bottom meter of the downcast and averaged over this meter for use in analysis as this reflected the closest point to the reef organisms. Additional physical data was retrieved from the Caribbean Ocean Observing System St. John Oceanographic Buoy (VI 104; 18°15.09′ N, 64°46.02′ W; https://www.caricoos.org/station/st-john/us, accessed on 22 March 2021).

#### 4.2.2. Benthic Temperatures

Benthic temperatures were taken at each site by a shaded Hobo Water Temperature Pro V2, Onset Computer Corp., Bourne, MA, USA) affixed to a steel rod within 20 cm of the reef surface following prior methods [69,71]. Probes were calibration-checked prior to deployment in an ice bath and took readings every 15 min. Data were averaged over each day to determine 1-day mean, and further averaged over 7, 14, 21, and 30 days for the respective means for use in analyses.

#### 4.2.3. Precipitation

Precipitation and wind data for the region was recorded at the St. Thomas Cyril E. King Airport by a US National Weather Service station (TIST) and data was accessed at the National Climate Data Center https://www.ncdc.noaa.gov/cdo-web/datasets/GHCND/stations/GHCND:VQW00011640/detail, accessed on 22 March 2021). The mean daily precipitation for the 14 days prior to a sampling event was calculated from daily summaries.

#### 4.2.4. Nutrient Analyses

Water samples for nutrient analyses were collected in whirlpak bags and stored on ice until return to the University of the Virgin Islands (UVI; within 5 h). Once at the laboratory, samples were transferred to acid-washed, sample-rinsed polypropylene bottles and frozen at −20 °C. Samples were shipped frozen to Woods Hole Oceanographic Institute (WHOI) and analyzed for inorganic nitrate plus nitrite (hereafter termed ‘‘nitrate’’), ammonium, silicate, and phosphate using a Lachat Instruments QuickChem 800 four-channel continuous flow injection system. This method is USEPA approved for nutrient analysis ranging from groundwater to the open ocean.

#### 4.2.5. Benthic Community Composition

Benthic cover at each study site was estimated using digital video along six randomly sited permanent transects as described in [71]. Each transect was 10 m in length and marked with steel rods, with transects spaced at least 3 m apart. Digital video was recorded perpendicular to the substrate and resultant images were cut into non-overlapping images, typically 15 per transect. Fifteen random points were placed on the image using Coral Point Count software [72] and characterized to the lowest identifiable taxonomic or abiotic level by a trained expert. Cover of each category (i.e., Coral, Gorgonians, Sponges, Zoanthids, Macroalgae, Coralline Algae, Dead Coral with Turf Algae, Non-Living Substrate, Other Living) was calculated for each transect by dividing the number of occurrences by the total number of points surveyed. Macroalgal cover was partitioned into % fleshy macroalgae, % *Dictyota* spp., % *Lobophora variegata*, % *Halimeda* spp., and % other.

### 4.3. Biological Sampling

#### 4.3.1. Collection of *Gambierdiscus* Epiphytes

*Gambierdiscus* were collected as epiphytes on macroalgae to determine their abundance and toxin content and scientific collection permits for this project were approved by the Virgin Islands Department of Fish and Wildlife, Marine Resources Division.

*Dictyota* spp. were the most widely distributed algae at the sampling sites (and frequently was the only macroalgal taxa present), so only *Dictyota* spp. were sampled for this study. Eight replicate samples of *Dictyota* spp. (four for *Gambierdiscus* abundance and four for toxin measurements) were collected by SCUBA divers from each study site in each month of the study period, with these exceptions: February 2010—only Flat Cay was sampled, March 2010—only Coculus Rock and Seahorse Shoal were sampled, and September 2012—no sites were sampled. Multiple thalli of *Dictyota* spp. were collected by carefully cropping and transferring to a Ziploc bag, which was then sealed underwater. Samples were stored in a cooler until processing within the same day. For sample processing, macroalgae were vigorously shaken for at least one minute to loosen the dinoflagellates, which were then sieved sequentially using 200 µm and 20 µm nitex sieves. *Dictyota* spp. retained in the 200 µm filter were removed, blotted dry with a paper towel, and weighed. With samples for *Gambierdiscus* abundance, the fraction of material retained on the 20 µm sieve was rinsed into a 15 mL conical tube, brought up to 10 mL with filtered seawater, and preserved with 0.5 mL formalin. For toxin samples, material retained on the 20 µm sieve was pooled from all four samples and rinsed into a shallow tray. The tray was maintained under low light (cool white fluorescent) and larger particulate material was allowed to settle to the bottom of the tray while living, detached *Gambierdiscus* cells (or other motile epiphytes) would remain in the overlying water due to active swimming and phototaxis. This overlying water was gently siphoned off and sieved again with a 20 µm sieve. Material collected with this final sieving was rinsed into 50 mL polypropylene centrifuge tubes (total capacity 60 mL) with filtered seawater and samples centrifuged at low speed (<1000× *g*) for 5 min to pellet cells. A small volume of overlying seawater was discarded to reach a final volume of 50 mL for each sample. This sample was then inverted several times to mix and a 1 mL aliquot was collected and preserved as described above for *Gambierdiscus* abundance measurements to determine the *Gambierdiscus* cell density within the toxin sample. After low-speed centrifugation of the remaining cell suspension and subsequent removal of supernatant, the cell pellet was stored at −20 °C or on dry ice (during shipping to Dauphin Island) prior to toxin analyses.

#### 4.3.2. *Gambierdiscus* Cell Enumeration

Preserved samples were gently shaken and 0.5–1.0 mL was loaded in a Sedgewick Rafter slide. *Gambierdiscus* cells were identified to genus based on cell size and shape using photomicrographs and line drawings, e.g., [73]. *Gambierdiscus* abundance was enumerated using a Zeiss Axioskop microscope at 100× magnification. Sample cell densities were determined by multiplying the summed cell counts by a subsample proportion factor, and then dividing this value by the *Dictyota* wet weight to express *Gambierdiscus* cell concentrations as cells g ww^−1^. At the time this study was conducted, methods for discriminating *Gambierdiscus* species had not yet been developed, so cell counts are given for total *Gambierdiscus* spp.

### 4.4. Toxin Extraction and Analyses

#### 4.4.1. Cell Pellet Extraction

Pellet material representing the 20–200 μm fraction of epiphytic material collected from *Dictyota* spp. (approx. 5 g per tube) were initially extracted in 10 mL of 100% methanol (MeOH) with 2 min. vortex mixing and 2 min. probe sonication on ice (5 s pulses, 20% power). The extract was centrifuged (3000× *g* for 5 min. at 20 °C) and the supernatant was collected. The sample pellet was then extracted two more times as before, but without additional probe sonication. Supernatants were pooled (30 mL total), diluted with water to 60% aqueous MeOH, and then partitioned three times with 25 mL dichloromethane (DCM). The recovered DCM fractions were pooled and dried by rotary evaporation at 30 °C. The sample residue was then quantitatively transferred from the evaporation flask with washes of MeOH and DCM, added to a 13 × 100 mm glass vial, and dried under high-purity nitrogen gas at 30 °C. Sample CTX residue was then redissolved in 100% MeOH (5 mL) with 2 min. vortex mixing and 2 min. bath sonication and stored at −20 °C until analysis. All extractions were performed with HPLC-MS grade solvents (Sigma) and ultra-pure (18 MΩ) water.

#### 4.4.2. Quantitation of CTX by In Vitro N2a Cytotoxicity Assay

The ciguatoxin content of each sample was measured as composite toxicity using an ouabain-veratridine (O/V) dependent in vitro neuroblastoma cytotoxicity assay (N2a assay) [74]. These assays utilized mouse Neuro-2a cells (ATCC, CL131; N2a), which were propagated and maintained under continuous growth as previously described [27,75]. Cells were harvested at 85–90% confluency and seeded to sterile 96-well polystyrene plates at a density of 4 × 10^4^ cells well^−1^. The N2a assay measures sample toxicity as a loss in viability of N2a cells that have been sensitized with O/V, making these cell responses highly specific to sodium channel toxins (e.g., CTX) and thus adds a line of evidence for CTX (or a composite of CTXs) being present in samples when loss in viability is observed. Within each assay, the response of untreated N2a cells (serving as negative control) is assessed at the same sample doses provided to O/V-treated cells to determine if the sample contains other toxic substances that are not sodium channel toxins and could affect viability of O/V-treated cells.

For quantitative assays of CTX, triplicate dose-response curves were determined for both O/V-treated and untreated N2a cells exposed to eight concentrations of sample CTX residues redissolved by high-speed vortex in 100 µL of 5%-FBS-RPMI media spanning a 128-fold range. After 24 h of exposure to sample extracts, N2a cell viability is assessed as the reduction of 3-(4,5- dimethylthiazol-2-yl)-2,5-diphenyltetrazolium bromide (MTT) by metabolically active cells to a purple formazan product that is measured by absorbance at 570 nm. The concentration of sample extract at which 50% of N2a cells lost viability (IC_50_) was compared to the IC_50_ of a purified C-CTX-1 standard (50 pg starting dose) that was measured in a concurrent N2a assay seeded with the same batch of N2a cells seeded to sample assays. These analyses were possible due to an aliquot of purified C-CTX-1 stock that was purified from toxic *Sphyraena barracuda* harvested from the Virgin Islands and is the same FDA stock reported in several prior studies [e.g., 27, 75, and others]. Impurities were assessed by LC-MS/MS analyses prior to use and original stocks quantified via NMR and gravimetric analysis (data not available). The toxin content of samples is expressed as the mass of C-CTX-1 equivalents in *Gambierdiscus* cells (pg C-CTX-1 eq. cell^−1^) based on the toxin sample cell counts described above. Samples in which the amount of extract required to cause a 50% loss in viability of O/V-treated N2a cells, and also caused significant loss in viability in untreated N2a cells, were considered below the limit of detection. Samples like these, in which there is clear CTX-specific activity, but quantitation criteria are not met, were considered positive detections of CTX, but their CTX content is described as “trace”.

#### 4.4.3. Sample Screening and Dose Determination

To determine which samples contained sufficient CTX-specific toxicity to be quantitated using the procedure described above, sample extracts were initially screened using eight sample concentrations along a two-fold dilution series that were assayed in triplicate. These sample concentrations ranged from 0.0078–1.0% of the total extract, corresponding to doses of ~0.7–1000 *Gambierdiscus* cells. The extract dissolution series was prepared in 100% MeOH, dried under high-purity N_2_ gas, redissolved in a minimum 100 µL of assay growth media, and 10 µL was added to assay well. Since a minimum of 100 µL was required to redissolve dried extracts, but only 10 µL of this could be used as a dose, a dosing range of 0.008–1.0% required using at least 10% of total sample extract. Due to the nature of determining IC_50_ values from sigmoidal dose-response curves that meet quality control criteria, full quantitative N2a assays require the highest doses to result in <20% N2a cell viability and be 8 to 16-fold higher than the IC_50_ and at least 32-fold higher than doses that show no CTX-specific toxicity. Thus, samples in which a dosing of 1% of the total extract (requiring 10% of the extract to be used) demonstrated CTX-specific toxicity, but failed to cause <20% N2a cell viability, would not contain sufficient sample material for a quantitative assay result requiring a higher dosage. These low-CTX samples were considered to be positive detections of CTX, but their CTX content is described was “trace” rather than a numerical value. Samples for which the maximum dose of 1% sample extract or subsequent dilutions caused CTX-specific toxicity resulting in <20% N2a viability were analyzed by full quantitative assays. The dose for these quantitative assays was adjusted to achieve the maximum concentration at which non-O/V-treated N2a cells maintain >90% viability and O/V-treated cells have <20% N2a viability and generate a dose response curve meeting quality criteria. Quantitation was based on the mean sample IC_50_ values of six replicate dose response curves (measured in two assay plates, each containing triplicate dose response curves) that showed less than 15% variation.

#### 4.4.4. Tests of Matrix Effects and Sample Purification

In many samples, the doses that could be used for quantitation were limited by their level of non CTX-specific toxicity in N2a cells, i.e., the doses required to produce a toxin response in O/V-treated N2a cells also caused a significant toxic response in untreated N2a cells. This reflects our use of MeOH extracts that received no purification beyond partitioning with DCM and thus would contain a variety of cell metabolites, such as free fatty acids, that could be toxic to N2a cells at high doses. Possible interference of matrix compounds and additional sample clean-up were tested on four quantifiable samples representing the range of toxin concentrations across extracts in this study. Analyses of a dilution series for each of these four samples showed a linear, proportional response to dilution in quantifiable concentrations and only 7–16% variation in determined CTX concentration, indicating a lack of matrix effects on measured CTX-specific toxicity. Solid-phase extraction (SPE) was performed on these samples using silica (Bond Elut Si; 100 mg; Agilent, Santa Clara, CA, USA). In all four samples, Si SPE purification caused a >40% reduction in toxicity to both O/V-treated and untreated N2a cells, indicating that CTX was being removed along with cytotoxic matrix compounds and that SPE clean-up would greatly affect accuracy of CTX measurement. Hence, further purification was not performed on any quantified samples to ensure accuracy rather than sensitivity. This decision was supported by C-CTX-1 spike recovery trials (below). The lowest concentration that could be quantified with confirmed CTX-specific toxicity was 0.28 ng C-CTX-1 eq. mL^−1^ (in extract), which represents the effective limit of quantitation for the samples in this study.

To better determine the limits of detection and possibility of matrix interference for N2a analyses of natural epiphyte assemblages, a sample containing no detectable CTX activity (as determined by the screening procedure described above) was spiked with C-CTX-1 standard at 8 concentrations ranging from 0.01–0.5 ng C-CTX-1 mL^−1^ (in extract) and the same range of concentrations were also produced in a dilution series (i.e., matrix concentration declined with CTX concentration). These tests indicated a limit of quantitation of 0.08 ng C-CTX-1 mL^−1^ in unpurified algal extracts and that CTX quantitation was not affected by dilution of sample matrix. However, this limit is not directly comparable to the detection limit of non-spiked samples since C-CTX-1 is a major component of bioaccumulated CTXs in fish, but has not yet been attributed as a major component in the toxin profiles of *Gambierdiscus* or *Fukoyoa* [76].

### 4.5. Statistical Analyses

*Gambierdiscus* abundance and toxicity data were tested for normality and homoscedasticity using a Shapiro–Wilk test and a Brown–Forsythe test, respectively. Log-transformed abundance data was parametric and mean abundance between sites and years were compared with one-way ANOVA followed by Tukey’s post hoc test for multiple comparisons. Toxicity data was highly skewed and non-parametric. A Box–Cox test was used to determine a transformation for toxicity data and all toxicity data was raised to the –2 power and mean values for sites and years were compared using a Welch’s ANOVA and a Games–Howell test for multiple comparisons. All univariate statistical analyses were performed in R. Benthic community composition across sites and years were compared using a cluster analysis and non-metric multidimensional scaling and their multivariate similarity was measured with an ANOSIM test, all performed using Primer-E. Multivariate correlations between benthic community composition or environmental conditions and *Gambierdiscus* abundance or toxicity were examined in Primer-E using the BEST routine. All reported multivariate results had a significance level of 0.1% (*p* < 0.001). Significance in the PCA analyses were based on the broken stick criterion of Peres-Neto et al. [77]. Graphs were created using GraphPad Prism version 9.0.0 for macOS (GraphPad Software, San Diego, California USA, www.graphpad.com, accessed on 9 June 2021).

## Figures and Tables

**Figure 1 toxins-13-00413-f001:**
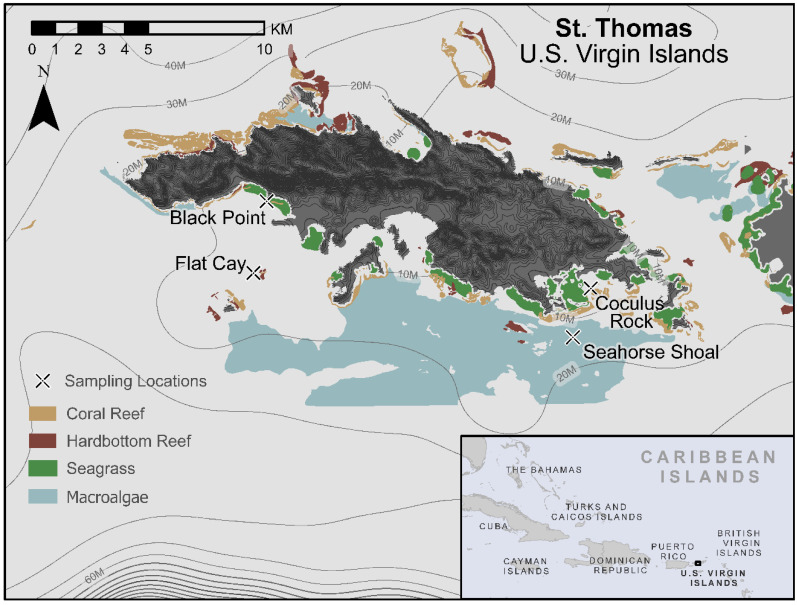
Map of long-term field sampling sites in St. Thomas, US Virgin Islands. Map created in ArcGIS Professional with overlaid shapefiles of benthic cover with coral, hardbottom, and seagrass from [35] and macroalgae from the St. Thomas and St. John benthic habitat dataset [36], both from the National Oceanic and Atmospheric Administration, U.S. Dept. Commerce.

**Figure 2 toxins-13-00413-f002:**
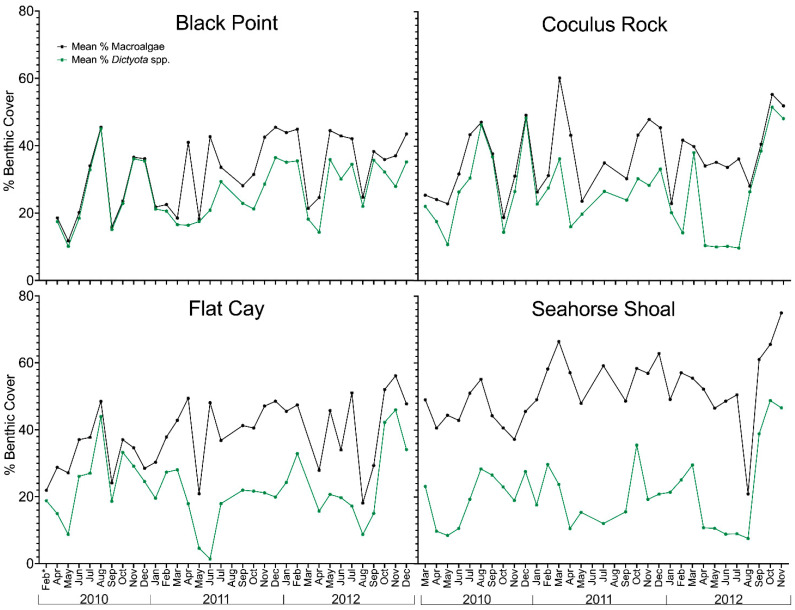
Time-series of benthic cover over the study period as determined by benthic habitat video surveys. Data highlights the temporal change in the percent (%) cover of combined macroalgae (black) and % *Dictyota* spp. cover (green) at each sampling site. Missing data points are time periods when benthic surveys were not conducted. Feb* denotes that FC was surveyed on 23 February 2010.

**Figure 3 toxins-13-00413-f003:**
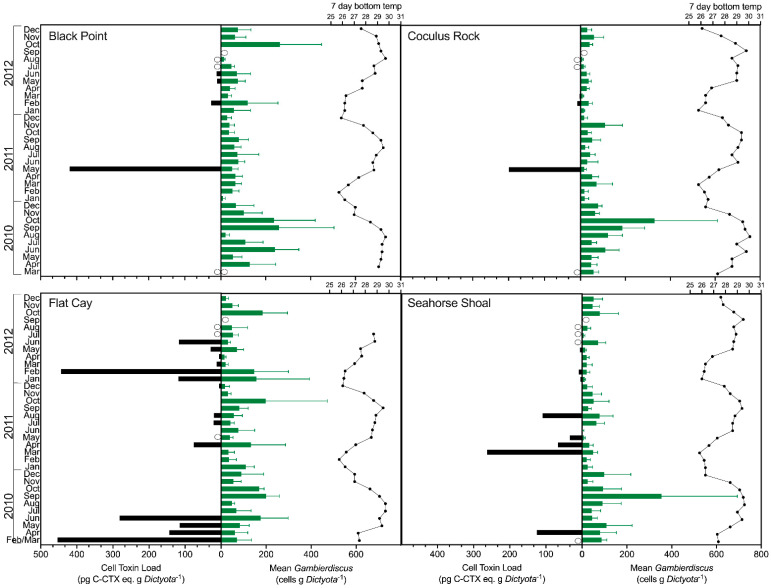
Asynchrony between cell toxin load (C-CTX eq cell^−1^ * cells g *Dictyota*^−1^) and mean (+s.d.) *Gambierdiscus* spp. abundance (cells g *Dictyota*^−1^), with 7-day averaged bottom temperatures from each site. Open circles represent “No Data” for either toxin load or *Gambierdiscus* abundance count.

**Figure 4 toxins-13-00413-f004:**
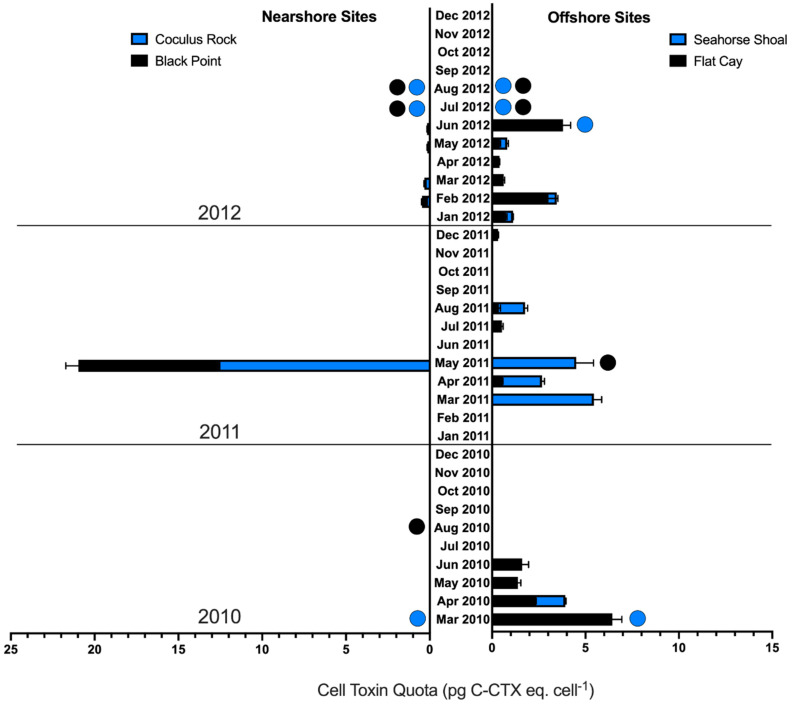
Time series of toxin cell quota (pg C-CTX-1 eq. cell^−1^) determined from benthic microalgal (20–200 µM fraction) field samples collected monthly from nearshore (Black Point; Coculus Rock) and offshore sites (Flat Cay; Seahorse Shoal) of St. Thomas, Virgin Islands. Black Point and Flat Cay (black bars) are western sites, whereas Coculus Rock and Seahorse Shoal (blue bars) are eastern sites. Colored circles represent “No Data” collected from the corresponding sites, with all other zeros indicating true non-detections of toxicity.

**Figure 5 toxins-13-00413-f005:**
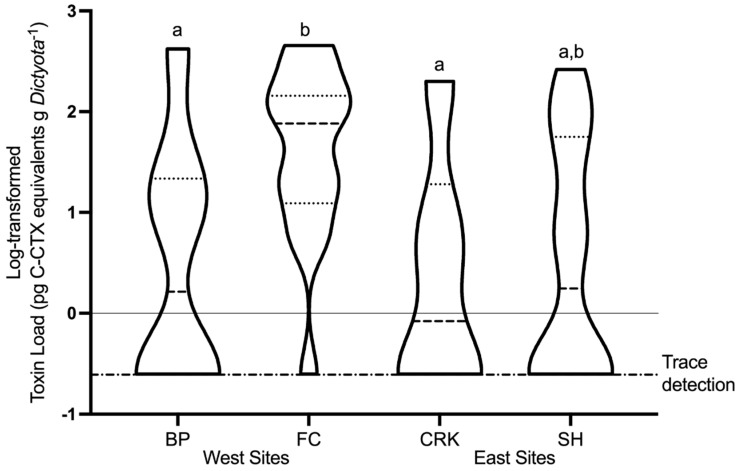
Truncated violin plot of log-transformed toxin load. Medians (dashed) and quartiles (dotted) shown. Trace detections were designated at toxin load 0.25 pg C-CTX-1 eq. g *Dictyota*^−1^ (log-transformed to −0.6). Letters on plot indicate significant differences in toxin load between sites.

**Table 1 toxins-13-00413-t001:** The distribution of positive and quantifiable samples as well as the mean and range of cell toxin quota and toxin load across all sites.

Field Site	Black Point	Flat Cay	Coculus Rock	Seahorse Shoal	Overall
Total Samples Collected	30	32	31	32	125
No. Positive Samples	8	15	7	16	46
No. Quantifiable Samples	4	14	3	9	30
% Total Positive Samples	17.4	32.6	15.2	34.8	% of total
% Total Quantifiable Samples	13.3	46.7	10	30	% of total
Mean Cell Toxin Quota (pg C-CTX-1 eq. cell^−1^)	0.3 ± 1.5	0.7 ± 1.4	0.4 ± 2.3	0.5 ± 1.3	0.5 ± 1.63
Cell Toxin Quota Range (pg C-CTX-1 eq. cell^−1^)	0–8.3	0–6.4	0–12.6	0–5.4	0–12.6
Mean Toxin Load (pg C-CTX-1 eq. g *Dictyota*^−1^)	15.1 ± 75.2	59.4 ± 121.3	6.8 ± 35.8	20.5 ± 55.5	25.5 ± 72.0
Toxin Load Range (pg C-CTX-1 eq. g *Dictyota*^−1^)	0–419.4	0–453.8	0–199.5	0–262.2	0–453.8

**Table 2 toxins-13-00413-t002:** Published values of in situ *Gambierdiscus* toxin quota. Toxin measurements determined by mouse bioassay (MBA) were originally compiled by Litaker et al. [15] and were standardized prior to conversion to toxin quotas, assuming one mouse unit = 18 ng of CTX3C and 72 ng of C-CTX-1 for Pacific and Caribbean samples, respectively [39,40].

Location	Cell Toxin Quota (pg CTX eq. Cell^−1^)		Method	*N*	Reference
	Range	Mean			
Northwest Hawaiian Is., Hawaii	-	24	^1^ MBA	1	[43]
Papara, Tahiti, French Polynesia	0.09–3.60	0.25 ± 0.18	^1^ MBA	34	[23]
Rapa Island, French Polynesia	0.5–13.5	-	^2^ RBA	4	[44]
Gambier Islands, French Polynesia	0.03–1.00	0.15 ± 0.26	^1^ MBA	6	[45]
Hitiaa Reef, Tahiti, French Polynesia	0.05–1.35	0.16 ± 0.20	^1^ MBA	10	[46]
Gambier Islands, French Polynesia	0.96–1.42	1.15 ± 0.32	^1^ MBA	2	[47]
Platypus Bay, Australia	-	0.23	^1^ MBA	1	[48]
Nuku Hiva, French Polynesia	0.85–3.90	2.38 ± 2.15	^2^ RBA		[49]
Rapa, French Polynesia	-	0.03 ± 0.004	^3^ N2a	1	[50]
^ St. Thomas, US Virgin Islands	1.14–5.14	1.54 ± 0.94	^1^ MBA	3	[51]
^ St. Thomas, US Virgin Islands	0.00–12.62	0.56 ± 1.75	^3^ N2a	125	This Study

^1^ Mouse Bioassay (MBA); ^2^ Radioligand Receptor Binding Assay (RBA), ^3^ In vitro mouse neuroblastoma MTT based assay (N2a); ^4^ sample number (*N*). ^ Caribbean Region.

## Data Availability

Complete environmental and physical data associated with this work is provided for open access as a downloadable file. Complete toxin and *Gambierdiscus* abundance data is presented in this manuscript but is also available on request to the corresponding author.

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
