# Peer review of "Asynchrony of Gambierdiscus spp. Abundance and Toxicity in the U.S. Virgin Islands: Implications for Monitoring and Management of Ciguatera"

_toxins, 2021, doi:10.3390/toxins13060413_

Round 1
Reviewer 1 Report
This manuscript described cell density and CTX toxicity of Gambierdiscus sp. at St. Thomas, US Virgin Islands during 2010-2012. This observation might be interested in the scientists responsible for risk characterization, analysis, and management of CP in the Caribbean and other regions.
The manuscript was well written, but there seems to be some correction needed to accept and published in Toxins.
- The information of the reference C-CTX-1 was lack. Explain the detailed information of the reference toxin with reference (source, purity, how quantified the amount, etc.)
- Describe the distribution of Gambierdiscus species and toxicity in the Caribbean and also the sampling site in the introduction section
- Explain how each cell was identified as belonging to the genus Gambierdiscus.
Other comments are as follows:
L112
Is "°C" correct?
Figure 1
If the frequency of CP incidence is available within the island, describe it on the map to help understanding and considering your observations.
L141-142, 146
The genus and species should be in Italic
Figure 3
" HOBO" should be replaced by "bottom water" since "HOBO" is a product name.
"pg C-CTX eq cell-1 * cells g Dictyota-1" should be replaced by "pg C-CTX eq g Dictyota-1".
L193
"Silica SPE column (Agilent)" -> "Silica (Si) SPE column"
L196
It isn't easy to understand why the authors concluded as CTXs were removed with the Si SPE column. However, it could be understood that this treatment removed the cytotoxic matrix other than CTXs.
Table 1
In the line "Mean Cell Toxin Quota (pg C-CTX-1 eq. cell-1)"
"-" should be replaced by "±"
L300, 305 and other
"CTX-3C" should be "CTX3C"
L303
CBA-N2a -> cell-based assay-neuro blastoma cells 2A assay (CBA-N2a)
L304-309
CTX3C is produced by Gambierdiscus and oxidized to 51-hydroxyCTX3C, 2,3-dihydoxyCTX3C, and 2,3,51-hydroxyCTX3C and increasing toxicity via the food chain.
"a common Pacific congeners of CTX that is considered to be 10-fold more toxic than the Caribbean congener" is not correct. CTX1B is considered to be 10-hold more toxic than C-CTX-1.
CTX3C is considered less toxic than CTX1B. For example, EFSA adapted TEFs for CTX1B, CTX3C, and C-CTX-1 as 1, 0.2, and 0.1, respectively. (EFSA Journal 2010; 8(6):1627)
L308
CTX-3C should be CTX3C
L311
CTX-1 should be C-CTX-1
Table 2
Insert "French Polynesia" after "Tahiti".
"pg CTX eq. cell-1": it should be mentioned which analog of CTX is.
"N": explain what it is.
L411
In ref. 19, the authors used CTX3C as a reference, but they did not identify the toxic substance as CTX3C.
L494
Spell out "UVI", "WHOI", and "USEPA", and explain what they are if in need.
L537-539
I wonder how the authors concentrate the sample to be 50 mL using 50 mL tube.
L546
"The preserved sample"?
L554
4.4.1 Cell pellet extraction
Explain the approximate amount of the pellet, volumes of all solvents used (MeOH, DCM, water, etc.)
L555
"μM" should be "μm"?
L579
Explain how to measure the viability in detail.
MTT?
L592
Describe that the similarity of the dose-response curves of the samples and reference C-CTXs as one of the evidences the principal toxins (Nav selective cytotoxic substances) were suspected to be CTXs
L598
I wonder the residue (dried lipophilic extract) was dissolved in the growth media (water system). It might be suspended. Please describe it in detail. e.g., using sonication or vortex?
L631
It could be understood that the non-specific toxic substances were removed with Si SPE treatment. How do you think?
L637
Make sure that the concentration is in the sample solution that applied in the wells or t in the wells.
L645
What does the algal extract mean?
"0.08 ng C-CTX-1 mL-1" indicates the amount of toxin in the solution.
In this case, the concentration (LOD) in the extract should be express as per weight or cell.
If it means the concentration in the matrix-containing solution, the amount of the matrix in the solution should be explained.
L646-649
Is it sure that the Caribbean strains do not produce C-CTX-1?
Reviewer 2 Report
This is an excellent manuscript, encompassing a great deal of work. The introduction is excellent and contains a good summary of relevant previous work, including hypotheses regarding factors which may or may not correlate to CFP prevalence. I point this out because one of the significant things about this manuscript is how the authors show that CFP is NOT correlated to a number of environmental factors (at least for these study sites). The decoupling of Gambierdiscus abundance and toxin abundance is extremely important, and the conclusion that CFP is likely caused by the occurrence of a sparse population of a highly toxic species is critical to future monitoring programs.. The next level - integrating species of fish involved and their seasonal movements - is fertile ground for future work.
Round 2
Reviewer 1 Report
The manuscript became more clearly described by revision.
However, I have a few comments, as shown below.
As for the response to my comment, "L592 Describe that the similarity of the dose-response curves of the samples and reference CCTXs as one of the evidences the principal toxins (Nav selective cytotoxic substances) were suspected to be CTXs "
The authors respose: This was addressed on lines 574-76, so to avoid repetition we have expanded this sentence to clarify. “The N2a assay measures sample toxicity as a loss in viability of N2a cells that have been sensitized with O/V, making these cell responses highly specific to sodium channel toxins (e.g., CTX) and thus adds a line of evidence.
I understand the authors stated the toxicity they detected was Nav selective as similar as that of CTXs. The dose-response relationship is substance-specific. The different types of compounds show different dose-response curves, even if they have a similar mode of action. (e.g., TTX and STX share the same receptor site on Nav and provide similar symptoms to the patients, but the dose-response curves of them are different)
Mention that the dose-response curves of the reference toxin and the extract were similar to make sure the Nav selective toxin in the extracts were suspected to be CTXs. (To deny the existence of brevetoxins and other substances having similar toxic activity as CTXs).
Furthermore, show examples of the dose-response curves of the reference toxin and the extract in the supplementary Materials.
L605
"residues redissolved"
I understand that a very small amount of C-CTX-1 will dissolve in an aqueous solution.
Although I am not an English native speaker, I think "dissolved" is not suitable in this sentence.
Since the residue containing many insoluble substances in an aqueous solution, "suspended" might be suitable as a chemical term.
L615
Ref. 40 is a review paper. Is it correct?
L615-616
"Impurities were assessed by LC-MS/MS analyses prior to use and original stocks quantified via NMR and gravimetric analysis."
I think the reference material the authors used was only the reference material quantified by NMR. It may become a universal reference material of the C-CTX-1, and all studies on the C-CTX-1 carried out in the world will be traceable one reference material. Please describe detailed information of the quantitative NMR experiments of the reference material in the supplementary materials.
